# Development of a recombinase polymerase amplification lateral flow assay for the detection of active *Trypanosoma evansi* infections

Zeng Li[1,2], Joar Esteban Pinto Torres[1], Julie Goossens[1], Benoit Stijlemans[1,3], Yann G.-J. Sterckx[2◉‡], Stefan Magez[1,4,5◉‡*]

**1** Research Unit for Cellular and Molecular Immunology (CMIM), Vrije Universiteit Brussel (VUB), Brussels, Belgium, **2** Laboratory of Medical Biochemistry and the Infla-Med Centre of Excellence, University of Antwerp (UA), Campus Drie Eiken, Wilrijk, Belgium, **3** Laboratory of Myeloid Cell Immunology, VIB Center for Inflammation Research, Brussels, Belgium, **4** Laboratory for Biomedical Research, Ghent University Global Campus, Incheon, South Korea, **5** Department of Biochemistry and Microbiology, Ghent University, Ghent, Belgium

◉ These authors contributed equally to this work.
‡ These authors are joint senior authors on this work.
* stefan.magez@vub.be

## Abstract

### Background

Animal trypanosomosis caused by *Trypanosoma evansi* is known as "surra" and is a widespread neglected tropical disease affecting wild and domestic animals mainly in South America, the Middle East, North Africa and Asia. An essential necessity for *T. evansi* infection control is the availability of reliable and sensitive diagnostic tools. While DNA-based PCR detection techniques meet these criteria, most of them require well-trained and experienced users as well as a laboratory environment allowing correct protocol execution. As an alternative, we developed a recombinase polymerase amplification (RPA) test for Type A *T. evansi*. The technology uses an isothermal nucleic acid amplification approach that is simple, fast, cost-effective and is suitable for use in minimally equipped laboratories and even field settings.

### Methodology/Principle findings

An RPA assay targeting the *T. evansi* RoTat1.2 VSG gene was designed for the DNA-based detection of *T. evansi*. Comparing post-amplification visualization by agarose gel electrophoresis and a lateral flow (LF) format reveals that the latter displays a higher sensitivity. The RPA-LF assay is specific for RoTat1.2-expressing strains of *T. evansi* as it does not detect the genomic DNA of other trypanosomatids. Finally, experimental mouse infection trials demonstrate that the *T. evansi* specific RPA-LF can be employed as a test-of-cure tool.

**Data Availability Statement:** All relevant data are within the manuscript and its Supporting Information files.

**Funding:** This work was supported by a grant of the China Scholarship Council (CSC), a research grant of the University of Antwerp (DOCPRO1, FFB190197), a research grant of the Foundation for Scientific Research / Fonds voor Wetenschappelijk Onderzoek – Vlaanderen (G013518N) and a UGent BOF startkrediet (01N01518). This work was performed in frame of an Interuniversity Attraction Pole Program (PAI-IAP N. P7/41) and was supported by the Strategic Research Program (SRP3, VUB). BS was supported by the Strategic Research Program (SRP3 and SRP47, VUB). The funders had no role in study design, data collection and analysis, decision to publish, or preparation of the manuscript.

**Competing interests:** The authors have declared that no competing interests exist.

## Conclusions/Significance

Compared to other DNA-based parasite detection methods (such as PCR and LAMP), the *T. evansi* RPA-LF (*Tev*RPA-LF) described in this paper is an interesting alternative because of its simple read-out (user-friendly), short execution time (15 minutes), experimental sensitivity of 100 fg purified genomic *T. evansi* DNA, and ability to be carried out at a moderate, constant temperature (39˚C). Therefore, the *Tev*RPA-LF is an interesting tool for the detection of active *T. evansi* infections.

## Author summary

Neglected tropical diseases (NTDs) affecting humans and/or domestic animals severely impair the socio-economic development of endemic areas. One of these diseases, animal trypanosomosis, affects livestock and is caused by the parasites of the *Trypanosoma* genus. The most widespread causative agent of animal trypanosomosis is *T. evansi*, which is found in large parts of the world (Africa, Asia, South America, Middle East, and the Mediterranean). Proper control and treatment of the disease requires the availability of reliable and sensitive diagnostic tools. DNA-based detection techniques are powerful and versatile in the sense that they can be tailored to achieve a high specificity and usually allow the reliable detection of low amounts of parasite genetic material. However, many DNA-based methodologies (such as PCR) require trained staff and well-equipped laboratories, which is why the research community has actively investigated in developing amplification strategies that are simple, fast, cost-effective and are suitable for use in minimally equipped laboratories and field settings. In this paper, we describe the development of a diagnostic test under a dipstick format for the specific detection of *T. evansi*, based on a DNA amplification principle (Recombinase Polymerase Amplification aka RPA) that meets the above-mentioned criteria.

## Introduction

*Trypanosoma evansi* is a haemoflagellate parasite which is closely related to *T. brucei*, the causative agent of human sleeping sickness and nagana in animals [1]. *T. evansi* is the causative agent of "surra" or "mal de caderas", which is the most common and widespread trypanosomal disease of domestic and wild animals and is characterized by high morbidity and mortality. The parasite is mechanically transmitted by biting flies and is found in many regions around the globe [2–6]. Outbreaks of surra have been reported in all types of ungulates (camels, cattle, buffaloes, horses, pigs, and deer) in Africa [7], Asia [8–10], Latin America [11–13] and recently Europe [14–16]. While *T. evansi* is commonly known as non-infective to humans, human infections were recently reported and confirmed in India and Vietnam, indicating that *T. evansi* may be emerging as a potential human pathogen [17–20]. Control of *T. evansi* trypanosomosis is mainly accomplished by drug treatment, but resistance of *T. evansi* to trypanocidal compounds has been reported in Africa [21, 22] and in the far east of Asia [23].

*T. evansi* parasites are classified into two groups based on their kDNA minicircle type [24], which are characterised by the presence (Type A) or absence (Type B) of the gene encoding the RoTat1.2 variant surface glycoprotein (VSG) [25, 26]. *T. evansi* Type B are less commonly found and have only been reported to occur in certain regions in Africa [27–32]. In contrast,

*T. evansi* Type A are widespread. Many diagnostic methods are available to detect *T. evansi* infections and include parasitological, serological, and molecular assays [33]. While some methods detect both *T. evansi* Types A and B, others are specific to one of both types. Conventional blood smear examination technique is widely used in the field and detects both *T. evansi* Type A and B. However, it can only diagnose clinical stages of infection and not latent or chronic infection [34]. In addition, it is time consuming and requires both the presence of microscopy equipment and specifically trained personnel at the screening site. To overcome these shortcomings, the *T. evansi* card agglutination test (CATT/T. evansi) was developed. It is a standard test for epidemiological field studies of *T. evansi* Type A since it is based on the use of the *T. evansi* RoTat 1.2 VSG antigen as an agglutination agent for host antibodies [35]. The advantage of this technique is that it is fast, easy to execute and suitable for field diagnosis. The main disadvantage of the technique is the lack of discrimination between previous exposure and current infections. Indeed, the host antibodies that drive the reaction can be a result of an active infection, a past infection, repeated exposure without necessarily initiation of successful infection, or even polyclonal B cell activation by other infectious agents such as helminths [36].

The diagnosis of trypanosomosis has been improved by the development and application of DNA-based techniques such as PCR, which is a very sensitive and effective method for the detection of chronic infections or prepatent period of disease [37, 38]. The DNA of killed trypanosomes does not remain in the blood for more than 24 to 48 hours, thus PCR-based assays are highly suitable for the detection of active infections [39]. Several genes have been investigated as targets for the PCR-based diagnosis of *T. evansi*; these include the RoTat1.2 VSG gene (Type A specific) [40–42], ribosomal DNA [43], a region from r-RNA internal transcribed spacer 1 (ITS-1) [44], the gene encoding the invariant surface glycoprotein ISG-75 [45], and the VSG JN 2118Hu gene (Type B specific) [26, 28, 46, 47]. The drawback of PCR-based methods is that they require well-trained and experienced personnel and a laboratory environment suitable for correct protocol execution. Hence, they are difficult to deploy and maintain under most field conditions. An interesting alternative to PCR is the so-called Recombinase Polymerase Amplification (RPA) [48]. The reaction mechanism of RPA has been reviewed elsewhere [49, 50] and is summarized in Fig 1 (the figure legend contains a detailed explanation of the RPA reaction). This isothermal nucleic acid amplification technology is simple, fast, cost-effective and is suitable for minimally equipped laboratories as well as for use in the field [51]. Hence, RPA is especially useful in infectious disease diagnostics and epidemiological studies [52–55]. The RPA reaction can be completed in 10 to 20 minutes at temperatures between 24°C to 45°C [56]. The amplification product can be visualized by gel electrophoresis or in real-time by the inclusion of a nucleic acid dye. The specificity and sensitivity of RPA are typically enhanced by probe-based methods, which (depending on the type of probe) allow amplicon detection based on fluorescence or a lateral flow (LF) assay [48]. To date, RPA has been successfully applied for the detection of bacteria [57, 58], foodborne pathogens [59, 60], parasites [61, 62], and viruses [63, 64].

In this present study, we describe the development of the first recombinase polymerase amplification lateral flow assay for the detection of active Type A *T. evansi* infections (*Tev*RPA-LF). The *T. evansi* RoTat1.2 VSG gene was chosen as the target for the *Tev*RPA-LF for the following reasons: i) to ensure high specificity of the *Tev*RPA-LF for *T. evansi* as this parasite is closely related to *T. brucei*, ii) *T. evansi* Type A are most commonly encountered and widespread, and iii) to allow comparison with the previously described PCR targeting the *T. evansi* RoTat1.2 VSG gene [33]. We demonstrate that the *Tev*RPA-LF assay is highly specific for *T. evansi* since no cross-reactions with the closely related parasite *T. brucei* could be observed. In addition, we have tested the *Tev*RPA-LF in an experimental mouse model and demonstrate

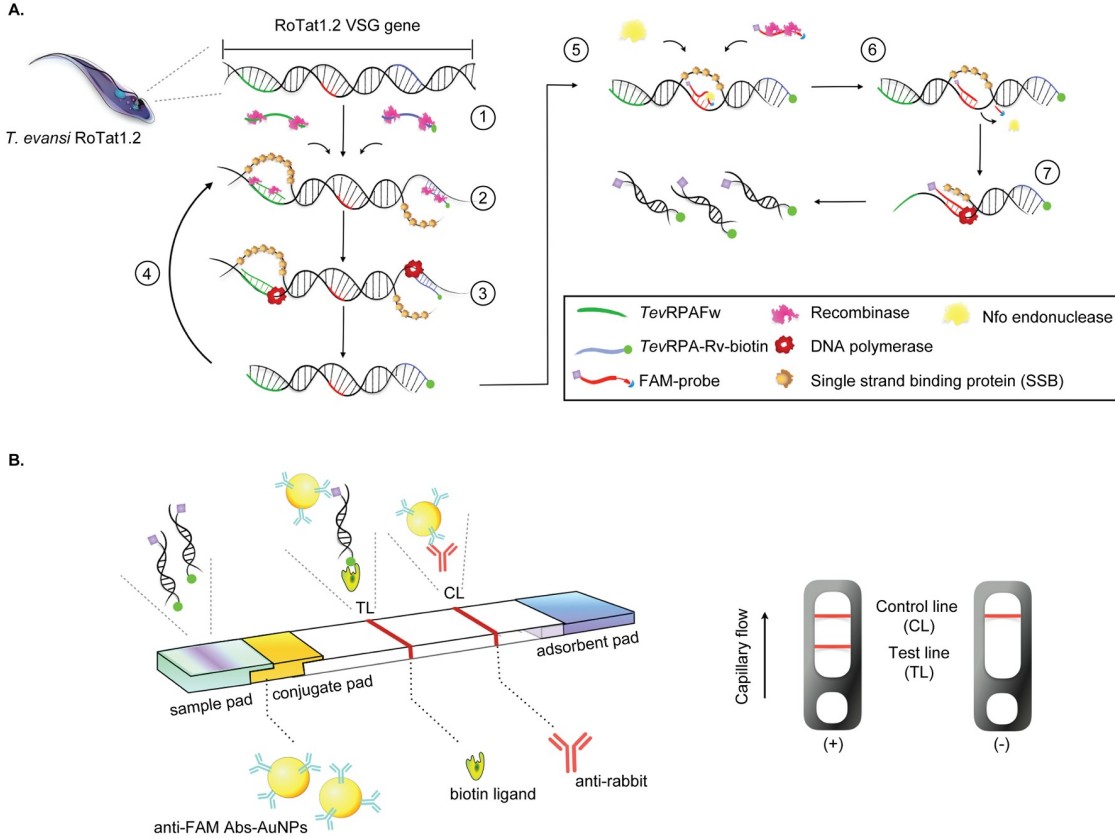

**Fig 1. Schematic representation of the *Tev*RPA-LF.** A: RPA-based generation of a *T. evansi* specific RoTat1.2 VSG amplicon for detection by a lateral flow (LF) assay. Step 1: two oligonucleotide primers (*Tev*RPA-Fw and *Tev*RPA-Rv-biotin) form a complex with the recombinase. Step 2: the primer-recombinase complexes invade the homologous sequences on the target DNA. Step 3: A DNA polymerase with a strand displacement activity performs amplification of the target sequence under isothermal conditions, resulting in the generation of a biotinylated amplicon. Step 4: the generated amplicons are again invaded by primer-recombinase complexes in a self-perpetuating cycle fueled in ATP by creatine kinase. Step 5: an oligonucleotide (FAM-probe) carrying a 5' FAM tag, a spacer sequence and a 3' blocking group forms a complex with the recombinase and invades the biotinylated amplicon generated in the previous steps. Step 6: only when the FAM-probe has successfully invaded the biotinylated amplicon and bound its complementary sequence, can the Nfo endonuclease bind and cleave the spacer region and 3' blocking group. Step 7: after removal of the 3' region of the FAM probe, the Nfo endonuclease dissociates. This allows the DNA polymerase to employ the cleaved FAM-probe as a forward primer. Together with the biotinylated reverse primer (*Tev*RPA-Rv-biotin) this leads to the formation of an amplicon bearing both the FAM and biotin tags. B: Read-out of the RPA via LF. The FAM- and biotin-tagged RPA product is mixed with the LF buffer, loaded onto the sample pad and is transported to the adsorbent pad through capillary flow. The RPA product is first bound by gold-labeled rabbit anti-FAM antibodies and later captured by a streptavidin-coated test line (TL). The control line (CL) is coated with anti-rabbit antibodies. While a valid negative test only contains a reddish band at the CL, a valid positive test will display bands at both the TL and CL.

that it can be used as a test-of-cure tool. The *Tev*RPA-LF described here has a processing time of 15 minutes and can be performed at a constant temperature of 39°C. Combined with the simplicity, robustness and reliability of the RPA-FL principle, the findings presented in this paper show that the *Tev*RPA-LF can be a promising tool for the detection of active *T. evansi* infections.

## Materials and methods

### Ethics statement

All experiments, maintenance and care of the mice complied with the European Convention for the Protection of Vertebrate Animals (ECPVA) used for Experimental and Other Scientific

**Table 1. Characteristics of trypanosomatid parasites used in this study.**

| Strain | Host | Country |
|---|---|---|
| *T. evansi* RoTat1.2 | Water buffalo | Indonesia |
| *T. evansi* STIB816 | Camel | China |
| *T. evansi* ITMAS180697 | Water buffalo | Vietnam |
| *T. evansi* 020499B | Horse | Columbia |
| *T. evansi* CAN86K | Dog | Brazil |
| *T. evansi* ITMAS060297 | Camel | Kazakhstan |
| *T. evansi* ITMAS050399C | Camel | Morocco |
| *T. congolense* Tc13 | Cow | Kenya |
| *T. vivax* TV700 | Cattle | Nigeria |
| *T. brucei* AnTat1.1 | Bushbuck | Uganda |
| *L. donovani* Ldl82 | Human | Ethiopia |

Purposes guidelines (CETS n° 123) and were approved by the Ethical Committee for Animal Experiments (ECAE) at the Vrije Universiteit Brussel (Permit Number: 14-220-31).

## Preparation of purified genomic DNA

Total genomic DNA of the different parasites used in this study (Table 1) was extracted and purified from infected mouse whole blood using a DNeasy Blood & Tissue Kit (Qiagen, Germany) according to the manufacturer's instructions. The DNA was eluted in 50 $\mu$l nuclease-free water and stored at -20°C until further use. The concentration and quality of the purified DNA were determined by gel electrophoresis (1% agarose gel run in TBE buffer at 110 V for 30 min) and spectrophotometric analysis (measurement of the absorbance at 260 nm, A260; examination of the ratio of the absorbances at 260 nm and 280 nm, $A_{260}/A_{280}$; performed on a NanoDrop-2000/2000c).

## Preparation of crude genomic DNA

Genomic DNA was robustly extracted by boiling. Briefly, 50 $\mu$l of blood was mixed with 10 $\mu$l nuclease-free water (Thermofisher). The sample was heated at 100°C for 5 minutes followed by centrifugation at 20000 g for 5 minutes, and the supernatant was applied as a crude DNA template. The DNA template was kept at -20°C until use.

## RPA primers and probes design

The primers and probes were manually designed based on the gene sequence of the Rode Trypanozoon antigenic type 1.2 VSG (RoTat 1.2 VSG) of *T. evansi* (GenBank accession code: AF317914.1). The NCBI's nucleotide BLAST tools combined with Primer 5 were used to search for primers specific to *T. evansi* without significant overlap with other genomes. The TwistAmp LF Probe oligonucleotide backbone includes a 5'-antigenic label FAM group, an internal abasic nucleotide analogue 'dSpacer' and a 3'-polymerase extension blocking group C3-spacer. The details of the primers and probes used are given in Table 2.

## Development and optimization of the *Tev*RPA assay

The RPA reactions were conducted with the TwistAmp Basic kit (TwistDx, Cambridge, UK). A 47.5 $\mu$l reaction mixture containing the following components was prepared in a 1.5 ml tube: 2.4 $\mu$l of both forward and reverse primers (final concentration: 480 nM), 29.5 $\mu$l

**Table 2. Primers and probes employed in this study.**

| Assay type | Primer name | Oligonucleotide (5'-3') | Reference |
|---|---|---|---|
| *Tev*RPA | *Tev*RPA-Fw<br>*Tev*RPA-Rv | CACCGAAGCAAGCGCAGCAAGAGGGTTAGCA<br>GTAGCTGTCTCCTGGGGCCGAGGTGTCATAG | This study |
| *Tev*RPA-LF | *Tev*RPA-Rv-biotin<br>FAM-Probe 1<br>FAM-Probe 2 | [Biotin]GTAGCTGTCTCCTGGGGCCGAGGTGTCATAG<br>[6F]TCTGCCCGCAGTTGCCTATGGCGGCGAAGT[dS]GCAGGGGCGATTTCAT[C3]<br>[6F]CTAAAATTTCTAAAGCACGCGGTTGGCAACA[dS]CAAGTTTGTGTGGGC[C3] | This study |
| PCR | RoTat1.2 Fw<br>RoTat1.2 Rv | GCGGGGTGTTTAAAGCAATA<br>ATTAGTGCTGCGTGTGTTCG | [40] |

6F stands for 6FAM, dS for dSpacer, and C3 for C3-spacer.

rehydration buffer supplied by the TwistAmp Basic kit, 12.2 $\mu$l nuclease-free water and 1 $\mu$l *T. evansi* purified genomic DNA (concentration of 120 ng $\mu l^{-1}$). The reaction mixture was then transferred to the kit's reaction tubes containing lyophilized enzyme pellet. Next, 2.5 $\mu$l magnesium acetate (MgAc; final concentration of 14 nM) was carefully pipetted onto the reaction tube lids. This was followed by a brief vortex and spin to mix MgAc with the RPA reaction mixture. The tubes were incubated in a thermocycler. To pinpoint the most optimal conditions for the *Tev*RPA, the samples were incubated at different reaction temperatures (25°C, 30°C, 35°C, 37°C, 39°C, 41°C, 43°C, 45°C, and 50°C) and for different durations (5 minutes, 10 minutes, 15 minutes, 20 minutes, 25 minutes, 30 minutes, 35 minutes and 40 minutes). Reactions were halted by placing the tubes on ice. The amplified products were first purified using the GenElute PCR Clean-Up kit (Sigma-Aldrich) and visualized on a 2% agarose gel.

## Development and optimization of the *Tev*RPA-LF

LF-RPA assays were performed following the indications provided in the TwistAmp nfo kit (TwistDx, Cambridge, UK). Briefly, the RPA reaction was assembled as described above (Materials and Methods subsection 'Development and optimization of the *Tev*RPA assay') with the exception of the addition of 2.1 $\mu$l of both forward and reverse primers (final concentration: 420 nM) and 0.6 $\mu$l probe (final concentration: 120 nM) to the reaction mixture. The amplified DNA was detected using LF strips (Milenia Hybridtech 1, TwistDx, Cambridge, UK) following the instructions indicated in the kit. Briefly, 1 $\mu$l of the amplified product was diluted with 99 $\mu$l LF buffer. Ten $\mu$l of this diluted sample was then loaded on the sample application area according to the manufacturer's instructions. The final result was visually read out after incubation for 2 minutes at room temperature. A testing sample was considered positive when both the detection line (biotin-ligand line) and the control line (anti-rabbit antibody line) were visible. A testing was considered negative when only the control line was visible (Fig 1). The amplicons could be analyzed on a 2% agarose gel after purification with the GenElute PCR Clean-Up kit (Sigma-Aldrich) to further confirm the testing result.

## Evaluation of sensitivity and specificity of the *Tev*RPA-LF

The specificity of the *Tev*RPA-LF was assessed by employing 20 ng of purified genomic DNA isolated from various parasites (Table 1). Samples containing only nuclease-free water were used as negative controls.

The sensitivity of the *Tev*RPA-LF was tested by employing the following concentrations of *T. evansi* purified genomic DNA as templates for the RPA reaction: 10 ng $\mu l^{-1}$, 1 ng $\mu l^{-1}$, 100 pg $\mu l^{-1}$, 10 pg $\mu l^{-1}$, 1 pg $\mu l^{-1}$, 100 fg $\mu l^{-1}$, 10 fg $\mu l^{-1}$ and 1 fg $\mu l^{-1}$. The results were analyzed by lateral flow and agarose gel electrophoresis.

## Comparison between *Tev*PCR and *Tev*RPA-LF in an experimental mouse infection model

C57BL6/C mice (bred in-house, 8 weeks old) were divided in two groups of six individuals. In each group, five mice were inoculated intraperitoneally with 2000 *T. evansi* (Rotat 1.2 strain) parasites in 200 $\mu$l of PSG buffer (36.4 mM NaCl, 3.12 mM NaH$_2$PO$_4$, 47.5 mM Na$_2$HPO$_4$ and 85.2 mM glucose, pH 8). The remaining mouse in each group was used as a negative control and was not infected. The mice were bled at different times post-infection. The mice in Group 1 were bled at days 1, 3, 5 and 6 post-infection. The animals in Group 2 were bled at days 0, 2, 4, 6, 8, 10 and 12 post-infection. All individuals from Group 2 were treated with Berenil (40 mg per kg), administered intraperitoneally at day 5 post-infection. For both groups, at each time point, 102.5 $\mu$l of whole blood was collected from the tail of each individual using nuclease-free tubes with 30 ml heparinized saline (10 units/ml; Sigma-Aldrich) to prevent coagulation. 2.5 $\mu$l of the collected blood was used to follow-up mice parasitemia by diluting the sample 200-fold (during high parasitemia periods) and 100-fold (during low parasitemia periods) in PSG buffer and counting the parasites under the light microscope. The rest of the collected blood (100 $\mu$l) was split into two parts to evaluate the samples using the *Tev*PCR and *Tev*RPA-LF. Fifty $\mu$l of collected blood was employed to prepare purified genomic DNA for the *Tev*PCR, whereas the remaining 50 $\mu$l of collected blood was used to obtain crude genomic DNA for the *Tev*RPA-LF. The *Tev*PCR was performed as described in [40] with the following modifications: the amount of purified genomic DNA as starting material (250 ng vs. 3000 ng) and the addition of 10% DMSO to the reaction mixture.

## Results and discussion

### Development and optimization of the *Tev*RPA

The first requirement of the *Tev*RPA-LF is a high specificity for the detection of *T. evansi*. This parasite is closely related to *T. brucei* and thus the selection of an appropriate nucleotide sequence that is unique to *T. evansi* is crucial. This is the case for a specific region (bp 1 to bp 1300) of the *T. evansi* RoTat1.2 VSG gene [40–42], which forms the target of the *Tev*RPA-LF for *T. evansi* detection (Fig 1). This limits the use of the *Tev*RPA-LF described here to the detection of Type A *T. evansi*, and not Type B. Based on this particular region, a primer pair was designed for the *Tev*RPA such that the resulting amplicon does not exceed 500 bp (as suggested by the RPA manufacturer instructions). As can be seen from Fig 2A, an RPA with this primer pair (initially incubated at 37˚C for 30 minutes) on *T. evansi* purified genomic DNA extracted from infected mice blood yields an amplicon of around 289 bp. The reaction was also performed on genomic DNA purified from a naive mouse to exclude the possible lack of specificity due to cross-reactivity. No amplification could be observed in this negative control sample (Fig 2A).

Next, the assay conditions were optimized by allowing the RPA reaction to proceed at various incubation temperatures and amplification times. First, a range of incubation temperatures between 25˚C and 50˚C were tested at a constant amplification time of 30 minutes. As can be seen from Fig 2B, 39˚C represents the most optimal incubation temperature as it produces the highest amount of amplicon. In a second phase, the RPA was performed at a constant incubation temperature of 39˚C while varying the amplification times from 5 to 40 minutes in 5 minute increments (Fig 2C). Although the *Tev*RPA can be performed within 10 minutes, longer incubation times clearly yield a higher signal. The amplification time of 15 minutes was selected in an effort to maintain a balance between providing maximum sensitivity and obtaining a minimal reaction time. In conclusion, these experiments demonstrate that

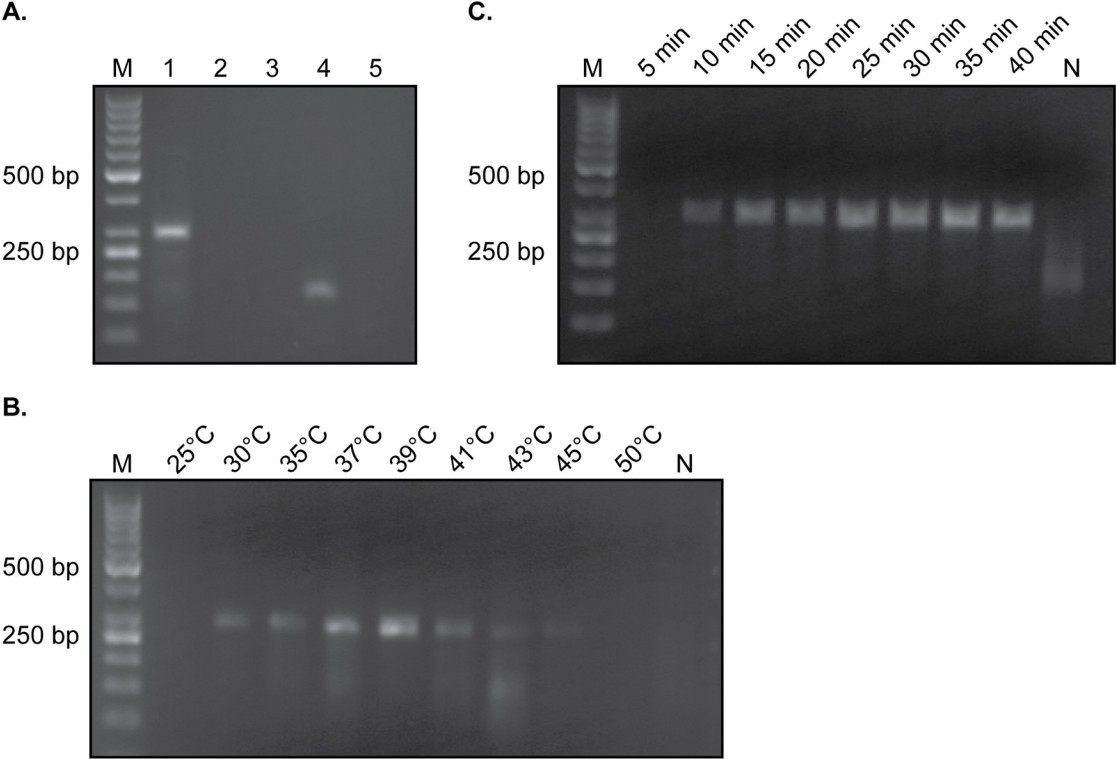

**Fig 2. Optimization of the *Tev*RPA.** A: Initial RPA incubated at 37˚C for 30 minutes on various samples. *Lane 1*, *T. evansi* purified genomic DNA; *Lane 2*, naïve mouse purified genomic DNA; *Lane 3*, sample without any template; *Lane 4*, RPA kit positive control; *Lane 5*, RPA kit negative control. B: RPA reaction on *T. evansi* purified genomic DNA incubated at different temperatures for a constant time of 30 minutes. C: RPA reaction on *T. evansi* purified genomic DNA incubated at a constant temperature of 39˚C for various times. In all panels *Lane M* indicates the molecular mass marker, whereas *Lane N* in panels B and C represents a negative control sample (no template DNA).

the TevRPA may be reliably performed with an amplification time of 15 minutes and an incubation temperature of 39˚C. These conditions were maintained for all subsequent experiments.

## The *Tev*RPA can be translated into a specific and sensitive *Tev*RPA-LF

The visualization of the RPA amplicon via agarose gel electrophoresis requires an additional purification step to avoid smeared bands on the gel due to the presence of enzymes and crowding agents [50]. This additional handling step is not necessary if the assay's read-out is performed via a lateral flow (LF) device [48, 49]. However, the translation of an RPA to an RPA-LF necessitates the addition of a labeled probe to the RPA reaction mixture and the biotinylation of the RPA reverse primer (Fig 1). Two candidate probes were screened for their potential to generate an RPA-LF for *T. evansi* detection (from here on referred to as *Tev*RPA-LF). Although both probes gave rise to positive signals when tested on *T. evansi* purified genomic DNA in both agarose gel electrophoresis and lateral flow detection formats, probe 1 clearly generates false positives while probe 2 does not (Fig 3A, right and left panels, respectively). Therefore, probe 2 was selected to be incorporated in the RPA assay to allow post-amplification detection of the amplicon via the TevRPA-LF.

Next, the specificity of the TevRPA-LF was evaluated by employing purified genomic DNA of various *Trypanosoma* and one *Leishmania* species as starting material for the amplification

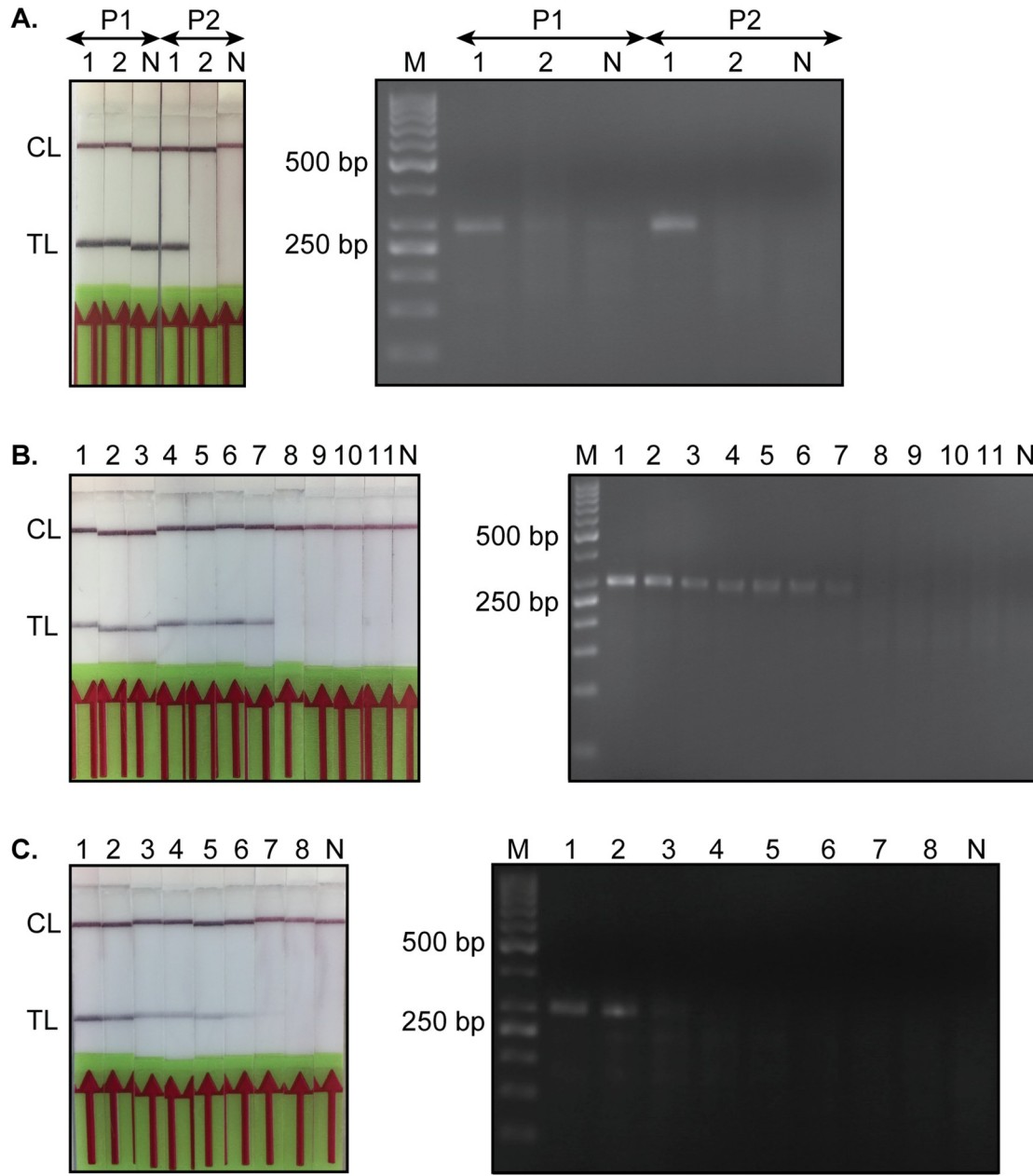

**Fig 3. Read-out of the *Tev*RPA via a lateral flow assay (*Tev*RPA-LF) and agarose gel electrophoresis.** A: Selection of a suitable probe for the development of the *Tev*RPA-LF. P1 and P2 refer to FAM probes 1 and 2, respectively. *Lane 1*, *T. evansi* purified genomic DNA; *Lane 2*, naïve mouse purified genomic DNA. B: Assessment of the specificity of the *Tev*RPA-LF. *Lanes 1-7*, various *T. evansi* strains as listed in Table 1; *Lane 8*, *T. congolense*; Lane 9, *T. vivax*; Lane 10, *T. brucei*; Lane 11, *L. donovani*. C: Comparison of the sensitivities of the *Tev*RPA by a lateral flow assay and agarose gel electrophoresis. *Lanes 1-8*, 10-fold dilution series of *T. evansi* purified genomic DNA starting at 10 ng $\mu l^{-1}$ (1 $\mu$l was loaded onto the gel). *Lane 1*, 10 ng; *Lane 2*, 1 ng; *Lane 3*, 100 pg; *Lane 4*, 10 pg; *Lane 5*, 1 pg; *Lane 6*, 100 fg; *Lane 7*, 10 fg; *Lane 8*,1 fg. All panels display the read-out of the *Tev*RPA by a lateral flow assay (left) and agarose gel electrophoresis (right). In all panels *Lane M* indicates the molecular mass marker, whereas *Lane N* represents a negative control sample (no template DNA). CL and TL refer to the control and test lines, respectively.

reaction. Only *T. evansi* genomic DNA resulted in visible bands at the test line, while the genomic material of other trypanosomatids did not result in any detection (Fig 3B).

Finally, the detection limit of the *Tev*RPA-LF was compared to the sensitivity of amplicon visualization via agarose gel electrophoresis by performing the *Tev*RPA on a 10-fold dilution

series ranging from 10 ng to 1 fg *T. evansi* purified genomic DNA per reaction (Fig 3C). When visualized using agarose gel electrophoresis, the lowest amount of genomic DNA that produces an amplicon that can be detected is 100 pg. In contrast, the *Tev*RPA-LF allows amplicon detection at an amount of 100 fg genomic DNA, which is 1000-fold more sensitive compared to agarose gel electrophoresis. The loss of sensitivity during post-amplification visualization via agarose gel electrophoresis is most probably related to the additional required purification step [65]. Hence, for the *Tev*RPA, the extra purification step comes at the cost of sensitivity, which advocates the use of the *Tev*RPA-LF over the *Tev*RPA followed by agarose gel electrophoresis.

## The *Tev*RPA-LF can detect active *T. evansi* infections in an experimental mouse model

Next, the *Tev*RPA-LF was evaluated for its potential to differentiate between ongoing and past infections in an experimental mouse model. In this experiment, C57BL/6 mice infected with T. evansi RoTat1.2 were divided into two groups and the presence of parasites was analyzed by microscopy, the previously described *Tev*PCR [40] and the *Tev*RPA-LF at various time points. Group 1 was left untreated, while Group 2 was treated with Berenil at 5 days post-infection.

As shown in Figs 4 and 5, all three techniques yielded identical results for most of the collected samples. A discrepancy between the detection methods was only observed at 3 days post-infection in Group 1; while parasites could only be detected in 3 out of 5 mice by microscopy, all samples were found to be positive when tested by the *Tev*PCR and *Tev*RPA-LF (Figs 4A and 5A). It is noteworthy to mention that in Group 1 only 4 samples from infected mice were available for testing at day 6 post-infection due to the premature death of one mouse. As expected, all infected mice in Group 1 succumbed to the infection at 7 days post-infection. In contrast, the mice in Group 2 survived day 7 post-infection indicating successful parasite clearance after Berenil treatment at day 5 post-infection. One mouse in Group 2 did not display

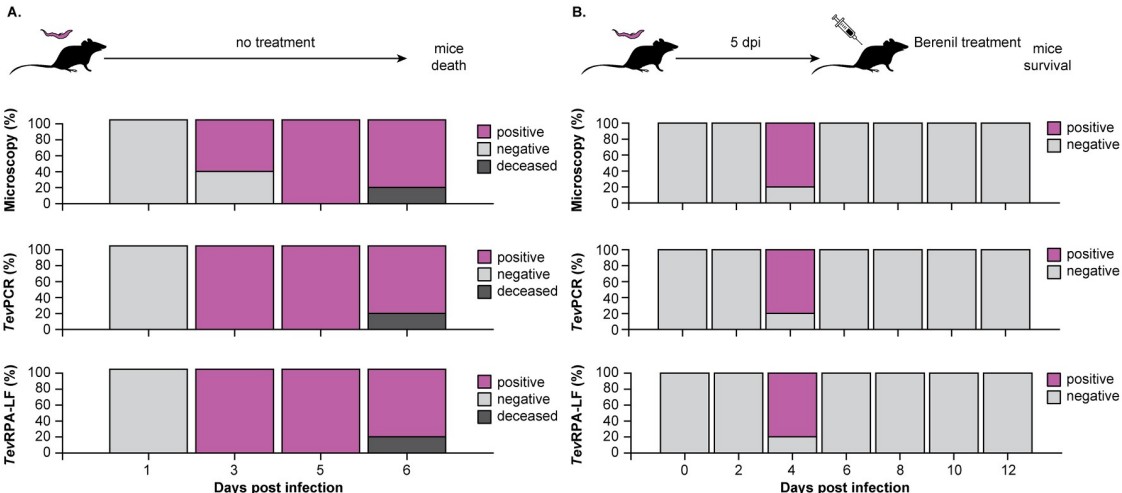

**Fig 4. Evaluation of the *Tev*RPA-LF as a test-of-cure tool in *T. evansi* infections in mice.** A: C57BL/6 mice were infected with *T. evansi* RoTat1.2 (n = 5) and the presence of parasites was monitored over the course of the infection by microscopy (top panel), the *Tev*PCR (middle panel, performed on parasite genomic DNA purified from the collected blood samples), and *Tev*RPA-LF (bottom panel, executed on crude parasite genomic DNA extracted from the collected blood). The results are displayed as the percentages of mice that scored positive or negative as determined by the above-mentioned techniques. B: C57BL/6 mice infected with *T. evansi* RoTat1.2 (n = 5) were treated with Berenil at 5 days post-infection. The presence of parasites was followed by microscopy, the *Tev*PCR and the *Tev*RPA-LF throughout the experiment. The panels and color codes are the same as for panel A. The *Tev*PCR and *Tev*RPA-LF read-outs are shown in Fig 5.

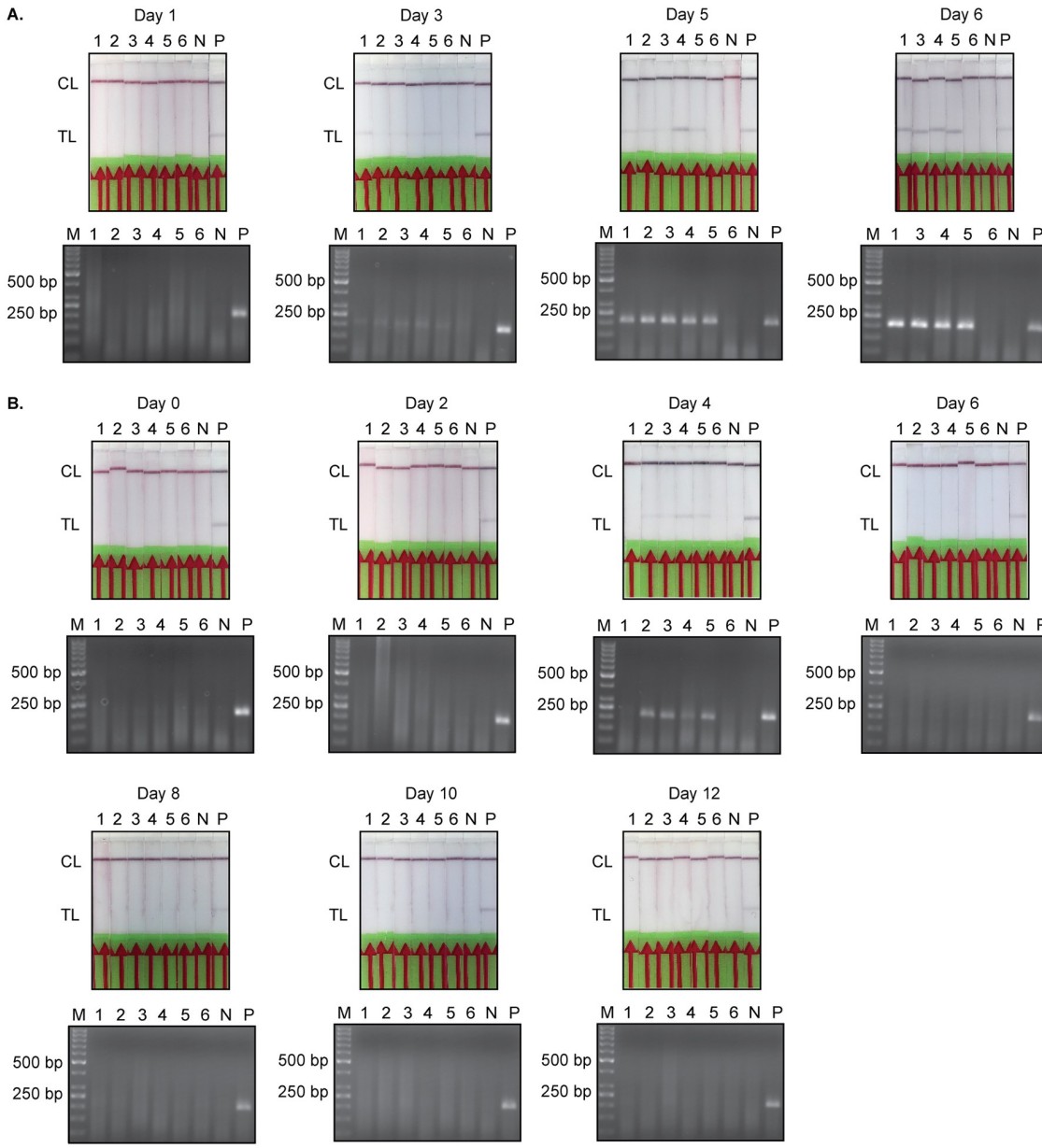

**Fig 5. *Tev*PCR and *Tev*RPA-LF read-outs.** The *Tev*PCR (bottom panels) and *Tev*RPA-LF (upper panels) read-outs displayed in Fig 4. A: *Tev*PCR and *Tev*RPA-LF results for the mouse infection trial of Group 1 mice (corresponds to the data set shown in Fig 4A). B: *Tev*PCR and *Tev*RPA-LF results for the mouse infection trial of Group 2 mice (corresponds to the data set shown in Fig 4B). In all panels *Lane M* indicates the molecular mass marker, *Lanes 1-6* indicate the individual mice (mouse 6 was used as a negative control within each data set and was not infected), *Lane N* is a negative control sample (no template DNA) and *Lane P* is the positive control (*T. evansi* purified genomic DNA). CL and TL refer to the control and test lines, respectively.

any signs of infection (4 days post-infection) and was scored as negative by all three methods. Importantly, no amplicons could be detected post-treatment by either the previously validated *Tev*PCR [40–42] or the *Tev*RPA-LF described in this work (Figs 4B and 5B). This demonstrates that the *Tev*RPA-LF is a suitable 'test-of-cure' assay. While both the *Tev*PCR and *Tev*RPA-LF display identical positive and negative score rates under these experimental conditions, the advantage of the *Tev*RPA-LF is that it is effective when performed with crude genomic

DNA, whereas execution of the *Tev*PCR requires additional purification of the isolated genomic DNA.

## Conclusion

*T. evansi* is the one of the most widespread causative agents of animal trypanosomosis in the world [6]. An essential part of parasite control is the availability of reliable, quick, and user-friendly diagnostic methods. In this paper, we have described the development of a *Tev*RPA-LF, a test that specifically detects active Type A *T. evansi* infections by amplifying a region in the *T. evansi* RoTat1.2 VSG gene. While the *T. evansi* RoTat1.2 VSG is also targeted by the *T. evansi* CATT [35] and *Tev*PCR [40–42] at the protein and DNA levels, respectively, the *Tev*RPA-LF presents some interesting advantages: i) compared to antibody-based tests (RoTat 1.2 CATT, Surra Sero K-Set, and *T. evansi* trypanolysis) the *Tev*RPA-LF can be employed to detect active parasitaemia and also serves as a test-of-cure tool since it is not hampered by the presence of infection-induced antibodies that could be the result of past infections or repeated parasite exposure without active infection and ii) the *Tev*RPA-LF combines the RPA format with a dipstick read-out, which outperforms a regular PCR in terms of user-friendliness and field applicability. While it can be argued that LAMP [66] offers the same advantage, the proposed LF format offers an advantage in terms of user friendliness as it visually resembles an antibody-test format that is already in place, while offering the advantage of detecting active infections. Based on the above-mentioned findings, the newly developed *Tev*RPA-LF presented in this paper provides a proof-of-concept with the potential of becoming a valid alternative for currently used screening tools. Its further development will require an additional evaluation of its performance in both experimental and clinical animal infection models.

## Acknowledgments

The authors wish to thank Prof. dr. Guy Caljon (LMPH, University of Antwerp) for providing samples of *L. donovani* genomic DNA.

## Author Contributions

**Conceptualization:** Zeng Li, Joar Esteban Pinto Torres, Yann G.-J. Sterckx.

**Data curation:** Zeng Li, Joar Esteban Pinto Torres, Yann G.-J. Sterckx.

**Formal analysis:** Zeng Li, Joar Esteban Pinto Torres, Yann G.-J. Sterckx.

**Funding acquisition:** Zeng Li, Stefan Magez.

**Investigation:** Zeng Li, Joar Esteban Pinto Torres, Julie Goossens, Benoit Stijlemans, Yann G.-J. Sterckx, Stefan Magez.

**Methodology:** Zeng Li, Joar Esteban Pinto Torres, Yann G.-J. Sterckx, Stefan Magez.

**Project administration:** Zeng Li, Joar Esteban Pinto Torres, Yann G.-J. Sterckx, Stefan Magez.

**Resources:** Zeng Li, Joar Esteban Pinto Torres, Julie Goossens, Benoit Stijlemans, Yann G.-J. Sterckx, Stefan Magez.

**Software:** Zeng Li, Joar Esteban Pinto Torres, Yann G.-J. Sterckx.

**Supervision:** Zeng Li, Joar Esteban Pinto Torres, Julie Goossens, Benoit Stijlemans, Yann G.-J. Sterckx, Stefan Magez.

**Validation:** Zeng Li, Joar Esteban Pinto Torres, Yann G.-J. Sterckx.

**Visualization:** Zeng Li, Joar Esteban Pinto Torres, Yann G.-J. Sterckx.

**Writing – original draft:** Zeng Li, Joar Esteban Pinto Torres, Yann G.-J. Sterckx.

**Writing – review & editing:** Zeng Li, Joar Esteban Pinto Torres, Julie Goossens, Benoit Stijlemans, Yann G.-J. Sterckx, Stefan Magez.

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
