## [Decision Letter · Decision Letter 0]

21 Oct 2019

Dear Dr. Magez:

Thank you very much for submitting your manuscript "Development of a Recombinase Polymerase Amplification Lateral Flow Assay for the Detection of Active Trypanosoma evansi Infections" (#PNTD-D-19-01398) for review by PLOS Neglected Tropical Diseases. Your manuscript was fully evaluated at the editorial level and by independent peer reviewers. The reviewers appreciated the attention to an important problem, but raised some substantial concerns about the manuscript as it currently stands. These issues must be addressed before we would be willing to consider a revised version of your study. We cannot, of course, promise publication at that time.

We therefore ask you to modify the manuscript according to the review recommendations before we can consider your manuscript for acceptance. Your revisions should address the specific points made by each reviewer. 

When you are ready to resubmit, please be prepared to upload the following:

(1) A letter containing a detailed list of your responses to the review comments and a description of the changes you have made in the manuscript.

(2) Two versions of the manuscript: one with either highlights or tracked changes denoting where the text has been changed (uploaded as a "Revised Article with Changes Highlighted" file); the other a clean version (uploaded as the article file).

(3) If available, a striking still image (a new image if one is available or an existing one from within your manuscript). If your manuscript is accepted for publication, this image may be featured on our website. Images should ideally be high resolution, eye-catching, single panel images; where one is available, please use 'add file' at the time of resubmission and select 'striking image' as the file type. 

Please provide a short caption, including credits, uploaded as a separate "Other" file. If your image is from someone other than yourself, please ensure that the artist has read and agreed to the terms and conditions of the Creative Commons Attribution License at http://journals.plos.org/plosntds/s/content-license (NOTE: we cannot publish copyrighted images). 

(4) If applicable, we encourage you to add a list of accession numbers/ID numbers for genes and proteins mentioned in the text (these should be listed as a paragraph at the end of the manuscript). You can supply accession numbers for any database, so long as the database is publicly accessible and stable. Examples include LocusLink and SwissProt.

(5) To enhance the reproducibility of your results, we recommend that you deposit your laboratory protocols in protocols.io, where a protocol can be assigned its own identifier (DOI) such that it can be cited independently in the future. For instructions see http://journals.plos.org/plosntds/s/submission-guidelines#loc-methods

While revising your submission, please upload your figure files to the Preflight Analysis and Conversion Engine (PACE) digital diagnostic tool, https://pacev2.apexcovantage.com/ PACE helps ensure that figures meet PLOS requirements. To use PACE, you must first register as a user. Then, login and navigate to the UPLOAD tab, where you will find detailed instructions on how to use the tool. If you encounter any issues or have any questions when using PACE, please email us at figures@plos.org.

We hope to receive your revised manuscript by Dec 20 2019 11:59PM. If you anticipate any delay in its return, we ask that you let us know the expected resubmission date by replying to this email.

To submit a revision, go to https://www.editorialmanager.com/pntd/ and log in as an Author. You will see a menu item call Submission Needing Revision. You will find your submission record there. 

Sincerely,

Rana Nagarkatti

Guest Editor

Alain Debrabant

Deputy Editor

Reviewer's Responses to Questions

**Key Review Criteria Required for Acceptance?**

**Methods**

-Are the objectives of the study clearly articulated with a clear testable hypothesis stated?

-Is the study design appropriate to address the stated objectives?

-Is the population clearly described and appropriate for the hypothesis being tested?

-Is the sample size sufficient to ensure adequate power to address the hypothesis being tested?

-Were correct statistical analysis used to support conclusions?

-Are there concerns about ethical or regulatory requirements being met?

Reviewer #1: (No Response)

Reviewer #2: (No Response)

Reviewer #3: 1. There is no discussion of a cutoff criterion that distinguishes a positive result from a negative result in the lateral flow detection. For example, in Figure 5A, Day 3 lateral flow results, the band for mouse #2 is not much more visible than mouse #6 (negative control), yet mouse #2 is labeled positive. My observations were made on the Figure5.tif, zooming in for a close-up and out for an overall view. Some method for objective evaluation of the band must be developed. Other strip assays build in a low positive and a high positive. A test band must be darker than the low positive and the high positive is included in an algorithm to assign a numerical value to the band intensity. A numerical score is not required for the TevRPA-LF, but objective scoring is.

2. This study involves the use of research animals (mice). Research with animals should be under the guidance of an Institutional Animal Care and Use Committee. I do not see where this is a requirement of PLOS NTD, however, the authors have not mentioned approval of an IACUC for the work performed, which I believe should be required. If the authors did the animal work under such supervision, please include a statement to that effect in the manuscript.

**Results**

-Does the analysis presented match the analysis plan?

-Are the results clearly and completely presented?

-Are the figures (Tables, Images) of sufficient quality for clarity?

Reviewer #1: (No Response)

Reviewer #2: (No Response)

Reviewer #3: The results are clear and well presented other than the issue stated above regarding objective criteria for negative versus positive scoring of the Lateral Flow test band.

**Conclusions**

-Are the conclusions supported by the data presented?

-Are the limitations of analysis clearly described?

-Do the authors discuss how these data can be helpful to advance our understanding of the topic under study?

-Is public health relevance addressed?

Reviewer #1: (No Response)

Reviewer #2: (No Response)

Reviewer #3: The authors offer accurate descriptions of the quality data they present.

The conclusion that the TevRPA-LF is a valid alternative for currently used screening tests overstates the evaluation that has been done. TevRPA-LF performed as well or better that alternatives in the few repetitions that were done. TevRPA-LF has promise to be easier to perform in the field than other available tests. However, the suggestion that it is ready to replace the other tests should come after more validation.

**Editorial and Data Presentation Modifications?**

Reviewer #1: (No Response)

Reviewer #2: (No Response)

Reviewer #3: 1. On page 6, line 125, the statement, “described previously in 2.5 with the exception of the addition of 2.1 μl of both forward” is not clear. The number 2.5 may refer to an earlier section that was numbered in an earlier draft. Please remove the number and replace with a proper reference to the place the information was described.

2. The description of RPA and its illustration in Figure 1 are incomplete and don’t fully explain how amplification occurs. The reference cited, #62, Daher et al. 2016, and the figure contained in that reference are much clearer. The explanation that the primers repeatedly re-invade the amplicons to generate new copies of the amplicon and the arrow on the figure in Daher 2016 indicating isothermal cycles of invasion and extension would help make the point in Figure 1 of the manuscript under review. 

3. On page 10, line 254, “the advantage of the TevRPA-LF is that it can be performed on crude genomic DNA samples.” A stronger statement could be made such as, “TevRPA-LF was effective when performed with crude genomic DNA" because this is in the results section. A “can be performed” statement would be appropriate in the Conclusion.

**Summary and General Comments**

Reviewer #1: This paper describes the development of a new diagnostic test for Trypanosoma evansi based on Recombinase Polymerase Amplification technology. The RPA test is an alternative to existing DNA-based PCR or LAMP diagnostic tests. The work is clearly explained on the whole. The limitations of the test in terms of sensitivity and ability to detect all T. evansi need to be included. The explanation of RPA methodology needs to be improved for clarity.

Abstract – the sensitivity of the RPA test could be mentioned.

I believe that Type A and B T. evansi were named after their kDNA minicircle type and therefore the PCR diagnostic tests based on minicircles should be mentioned.

On p3 line 49 – some explanation of how RPA works would be useful at this point for readers unfamiliar with the technique.

Table 1 – the host is given as “cattle” which is a plural term and out of line with the other host terms used. Use cow instead.

Line 125 “in 2.5 with the exception of the addition of…”. Meaning needs to be clarified.

Line 144 and 221 - useful to give the equivalent number of parasites.

Line 175 It should be pointed out the RoTat 1.2 gene is not present in all T. evansi – as in ref 27.

Line 184 Figure legend but no figure included – figures are at end with no list of figure legends to refer to.

Ref 38 seems incomplete.

Fig 1 is rather fussy and overcomplicated – it needs to be simplified, e.g. the trypanosome on the left is redundant, there is no need to show DNA as a double helix. Perhaps the figure could be split into RPA and RPA-LF?

Fig 5 is it necessary to show every mouse result?

Reviewer #2: The authors report the development of a novel molecular test for Trypanosoma evansi based on RPA combined with lateral flow detection and assessed the analytical sensitivity on serial dilutions of parasite DNA and its potential as test of cure on infected mice. While the study and data are technically sound, I have concerns and questions on the use of the developed the test in diagnosis and treatment of surra.

1. What is the added value of the test over T. evansi LAMP? Part of the authors published in 2018 in Vet Parasitol a study on RoTat 1.2 LAMP for sensitive T. evansi detection for diagnosis as well as test of cure (Tong et al. 2018). Why is this RPA needed and what is the added value over LAMP? LAMP amplification can be detected in real-time in simple closed-tube formats so what is the added value of a LF molecular test? 

2. The authors state that the test can be a valid alternative for the currently used screening tools. I’m not sure this is the case. How would the test fit in a diagnostic flow for surra given the already available tests RoTat 1.2 CATT, Surra Sero K-Set, T. evansi trypanolysis, PCR and LAMP? Where is the intended use of the tests? Currently Surra is rarely diagnosed in the field but at reference labs where molecular lab facilities are mostly available. This should be presented/discussed in the conclusions section.

3. Lines 31-33 on the low PPV of CATT, do the authors have any data or reference to support this statement?

4. Data on analytical sensitivity of RPA in serial diluted T. evansi DNA: this should be complemented by applying conventional T. evansi PCR (table 2) on the same serial dilutions. Same for the mice experiments with and without Berenil treatment.

5. How do the authors avoid sample contamination with PCR products from previous runs when applying later flow post-amplification, especially since the authors present the test for field use?

6. The study would be much stronger if a proof-of-concept can also be delivered on clinical samples.

Reviewer #3: The authors have assembled a test system for a disease of agricultural importance that is a particular problem in resource-limited areas of the world. The isothermal amplification and lateral flow detection are features that have potential to make the assay easier to perform in the field. The amount of characterization and validation performed is sufficient to consider this a proof of concept for the assay. As far as testing was done, the performance of the assay is good. At 100fg of DNA detected, the analytical limit is close to one parasite, which is good sensitivity. The authors have not tested clinical sensitivity in the sense of the lower limit of parasites in a blood sample from an infected animal. The infected animal tests are good, though limited in number and are taken from animals that would be expected to have a large parasite load. Further validation of the assay would require a larger number of animals over a range of infection conditions including lower parasite loads below the analytical limit of detection. Assaying infected animals after drug treatment is an important demonstration of the test’s ability to evaluate cure, a limitation of serological tests.

PLOS authors have the option to publish the peer review history of their article (what does this mean?). If published, this will include your full peer review and any attached files.

Reviewer #1: No

Reviewer #2: No

Reviewer #3: No

---

## [Decision Letter · Decision Letter 1]

9 Jan 2020

Dear Dr. Magez,

We are pleased to inform you that your manuscript, "Development of a Recombinase Polymerase Amplification Lateral Flow Assay for the Detection of Active Trypanosoma evansi Infections", has been editorially accepted for publication at PLOS Neglected Tropical Diseases.

Before your manuscript can be formally accepted and sent to production you will need to complete our formatting changes, which you will receive in a follow up email. Please note: your manuscript will not be scheduled for publication until you have made the required changes.

IMPORTANT NOTES

* Copyediting and Author Proofs: To ensure prompt publication, your manuscript will NOT be subject to detailed copyediting and you will NOT receive a typeset proof for review. The corresponding author will have one final opportunity to correct any errors when sent the requests mentioned above. Please review this version of your manuscript for any errors.

* If you or your institution will be preparing press materials for this manuscript, please inform our press team in advance at plosntds@plos.org. If you need to know your paper's publication date for media purposes, you must coordinate with our press team, and your manuscript will remain under a strict press embargo until the publication date and time. PLOS NTDs may choose to issue a press release for your article. If there is anything that the journal should know, please get in touch.

*Now that your manuscript has been provisionally accepted, please log into EM and update your profile. Go to http://www.editorialmanager.com/pntd, log in, and click on the "Update My Information" link at the top of the page. Please update your user information to ensure an efficient production and billing process.

*Note to LaTeX users only - Our staff will ask you to upload a TEX file in addition to the PDF before the paper can be sent to typesetting, so please carefully review our Latex Guidelines [http://www.plosntds.org/static/latexGuidelines.action] in the meantime.

Best regards,

Rana Nagarkatti, Ph.D.

Guest Editor

Alain Debrabant, Ph.D.

Deputy Editor

Reviewer's Responses to Questions

**Key Review Criteria Required for Acceptance?**

**Methods**

-Are the objectives of the study clearly articulated with a clear testable hypothesis stated?

-Is the study design appropriate to address the stated objectives?

-Is the population clearly described and appropriate for the hypothesis being tested?

-Is the sample size sufficient to ensure adequate power to address the hypothesis being tested?

-Were correct statistical analysis used to support conclusions?

-Are there concerns about ethical or regulatory requirements being met?

Reviewer #1: (No Response)

Reviewer #3: The authors have responded to all the Methods-related questions and comments in my original review. The changes, indicated by yellow highlight in the new draft of the manuscript are all acceptable.

**Results**

-Does the analysis presented match the analysis plan?

-Are the results clearly and completely presented?

-Are the figures (Tables, Images) of sufficient quality for clarity?

Reviewer #1: (No Response)

Reviewer #3: The authors have responded to all the Results-related questions and comments in my original review. The changes, indicated by yellow highlight in the new draft of the manuscript are acceptable.

The authors responded to my request that criteria be stated for the distinction between a positive result and a negative result. They suggest that this is a goal for future development of this assay. Their response indicates the preliminary nature of this proof-of-concept study.

**Conclusions**

-Are the conclusions supported by the data presented?

-Are the limitations of analysis clearly described?

-Do the authors discuss how these data can be helpful to advance our understanding of the topic under study?

-Is public health relevance addressed?

Reviewer #1: (No Response)

Reviewer #3: The authors have responded to all the Conclusions-related questions and comments in my original review. The changes, indicated by yellow highlight in the new draft of the manuscript are acceptable.

**Editorial and Data Presentation Modifications?**

Reviewer #1: (No Response)

Reviewer #3: The authors have responded to all the Editorial and Data Presentation Modifications-related questions and comments in my original review. The changes, indicated by yellow highlight in the new draft of the manuscript are acceptable.

**Summary and General Comments**

Reviewer #1: The authors have satisfactorily addressed all the issues raised in my review.

Reviewer #3: The authors have responded to all the Summary and general comments-related questions and comments in my original review. The changes, indicated by yellow highlight in the new draft of the manuscript are acceptable.

PLOS authors have the option to publish the peer review history of their article (what does this mean?). If published, this will include your full peer review and any attached files.

Reviewer #1: No

Reviewer #3: No

---

## [Editor Report · Acceptance letter]

10 Feb 2020

Dear Prof. Magez,

We are delighted to inform you that your manuscript, "Development of a Recombinase Polymerase Amplification Lateral Flow Assay for the Detection of Active Trypanosoma evansi Infections," has been formally accepted for publication in PLOS Neglected Tropical Diseases.

Best regards,

Serap Aksoy

Editor-in-Chief

Shaden Kamhawi

Editor-in-Chief
